# Patient experiences of an ankle fracture and the most important factors in their recovery: a qualitative interview study

Rebecca McKeown ![ORCID],[1] Rebecca Samantha Kearney ![ORCID],[1,2] Zi Heng Liew,[1,2] David R Ellard ![ORCID] [1]

¹Warwick Clinical Trials Unit, Warwick Medical School, University of Warwick, Coventry, UK
²University Hospitals Coventry and Warwickshire, Coventry, West Midlands, UK

**Correspondence to**
Rebecca McKeown;
R.McKeown.1@warwick.ac.uk

## ABSTRACT

**Objective** The objective of this qualitative research study is to explore patient experiences of ankle fracture and the factors most important to them in recovery.

**Design** Semistructured interviews exploring patient experiences of ankle fracture recovery at 16–23 weeks following injury. Interviews followed a topic guide and were recorded with an encrypted audio recorder and then transcribed verbatim. Thematic content analysis was used to identify themes in the data.

**Setting** Individuals were recruited from a sample of participants of a UK-based clinical trial of immobilisation methods for ankle fracture (ISRCTN15537280 at the pre-results stage at time of writing). Interviews were conducted at the participants' own homes or on a university campus setting.

**Participants** A purposive sample was used to account for key variables of age, gender and fracture management. Participants recruited from the clinical trial sample were adults aged 18 years or over with a closed ankle fracture.

**Results** Ten participants were interviewed, five of whom were female and six of whom needed an operation to fix their ankle fracture. The age range of participants was 21–75 years with a mean of 51.6 years. Eight themes emerged from the data during analysis; mobility, loss of independence, healthcare, psychological effects, social and family life, ankle symptoms, sleep disturbance and fatigue, and activities of daily living. Factors of importance to participants included regaining their independence, sleep quality and quantity, ability to drive, ability to walk without walking aids or weight-bearing restrictions, and radiological union.

**Conclusions** The results of this research demonstrates the extensive impact of ankle fracture on individuals' lives, including social and family life, sleep, their sense of independence and psychological well-being. The results of this study will enable an increased understanding of the factors of relevance to individuals with ankle fracture, allowing collection of appropriate outcomes in clinical studies for this condition. Ultimately these results will help formulate appropriate patient-centred rehabilitation plans for these patients.

**Trial registration number** ISRCTN15537280; Pre-results

## Strengths and limitations of this study

► Use of well-recognised reporting standards for purposes of transparency.
► Duplicate data analysis for consistency.
► Use of purposive sampling to account for key demographic variables.
► Participants were recruited from a clinical trial which restricted the timing of data collection.
► Individuals who declined to take part in the trial or the interview study may have had different experiences of recovery compared with those who agreed to participate.

demonstrates a bimodal distribution, usually affecting younger men and older women.[2 3] The incidence of ankle fracture is increasing and contributes to the rising economic cost of managing fractures in the current ageing population.[4] This cost of managing fractures in the UK is expected to reach £2.2 billion per annum by the year 2020.[5] While several clinical effectiveness trials have recently been published to ascertain the optimal management strategies for individuals with fractures of the lower limbs,[6–8] there is comparably less research into the patient experiences of recovering from these injuries. In 2018 a James Lind Alliance priority setting partnership on the subject of lower limb fractures in older people was completed and the sixth priority listed in this research area was 'what is most important to adults in their recovery from a fragility fracture of the lower limb?'.[9] This highlights the demand from academics and clinicians, and patients and members of the public for further research in this area.

The life impact of ankle fractures has been previously assessed in an article including interviews of patients and clinicians in the context of outcome measure development.[10] In this article we focus on patients only, with the aim of understanding their experience of ankle fracture recovery as well as the factors

## BACKGROUND

Ankle fractures are significant injuries which cause pain and reduced mobility.[1] The injury

most important to them. This will enable a greater understanding of the patient experience of recovery from this injury, to ensure that domains of interest to participants are being collected in the trials to assess clinical effectiveness of interventions for this injury. Furthermore, this will enable clinicians to achieve a broader knowledge base of the experiences of individuals with this injury and enable appropriate and effective patient-centred treatment plans to be formulated for these patients.

## Objectives

The objective of this qualitative study is to explore the patient experiences of ankle fracture and the factors most important to them during recovery.

## METHODS

This research was conducted in accordance with the Standards for Reporting Qualitative Research checklist.[11]

## Study design and methodological approach

We completed semistructured interviews with individuals who had sustained an ankle fracture at a single time point between 19 weeks and 23 weeks following injury. The qualitative approach used here was the thematic content analysis to focus on the participants' experience of their injury.[12] We took a realistic approach to the analysis, acknowledging that the individuals' ankle injuries exist in a reality outside of their own perception of it.[13]

## Participant identification

This study was embedded in the Ankle Injury Rehabilitation (AIR) Trial, an ongoing randomised controlled trial comparing plaster cast to functional brace in the treatment of individuals with an ankle fracture.[14] Participants of the trial were adults aged 18 years or over who had a closed ankle fracture either managed operatively or non-operatively. The eligibility criteria for the trial can be found in table 1.

## Sampling

We used purposive sampling for this research study. Participants who had previously stated that they were willing to be contacted for further research into ankle fractures were considered for invitation to the study. Participants were approached after completion of the 16-week questionnaire in the trial follow-up schedule as this was the primary outcome for the study. Interviews were completed after this time point to ensure that the interview did not influence the way in which participants answered the trial questionnaire. The sampling strategy allowed for a diverse range of patients with regard to their age, gender, fracture management (operative or non-operative) and allocated intervention within the trial (functional brace or plaster cast).

## Recruitment and consent

The participants of the trial were screened for sampling attributes of age, gender, fracture management and randomised intervention. We screened the online database of trial participants for the four sampling attributes mentioned above and invited a diverse range of individuals in relation to these attributes to participate. Invitation letters and participant information sheets were sent out to these individuals and the letter stated that we would telephone them in 1-week time to discuss the study. During the phone call, participants were given the opportunity to ask any questions. If they verbally consented to participate, a mutually convenient interview date and time was arranged between the interviewer and participant. At the interview consultation, the participant was given a further opportunity to ask questions. Once these had been answered satisfactorily they signed a consent form to confirm their willingness to participate. Participants were informed that they had the opportunity to withdraw their data at any time throughout the interview or up to 72 hours following the interview. We completed data analysis concurrently with the data collection so we could identify when no new themes were emerging from the data. We decided prior to commencing data collection

| Table 1 | Eligibility criteria of the Ankle Injury Rehabilitation (AIR) Trial |
| --- | --- |
| **Inclusion criteria** | **Exclusion criteria** |
| Able to give written informed consent | Ankle fracture secondary to known metastatic disease |
| Aged 18 years or over | Complex intra-articular fracture (eg, pilon fracture) |
| A closed ankle fracture managed operatively or non-operatively for which the treating clinician would consider plaster cast a reasonable management option | In the opinion of the surgeon the patient would require manipulation and close contact/moulded casting |
| Randomised within 3 weeks of operative management or injury if non-operative | Wound complications contraindicating functional brace intervention |
| | Previous ankle fracture already randomised in the present trial |
| | Known pre-existing neuropathic joint disease contraindicating functional brace intervention |
| | Unable to adhere to trial procedures or complete postal questionnaires |

**Box 1  Example questions used during the interviews**

Example questions used in interviews

Could you explain how your ankle fracture has impacted your day-to-day life?

How has your ankle fracture affected your walking?

Could you talk to me about the impact of your ankle fracture on your family life?

Could you explain what was most important to you when recovering from your ankle fracture?

What bothered you most throughout your recovery from your injury?

You mentioned that…was an important factor to you. Could you tell me more about that?

Did your ankle fracture affect your mood in any way?

How did your ankle fracture affect your work?

How did your ankle fracture affect your leisure activities or use of free time?

and analysis that data collection would be stopped when we reached the point where two consecutive interviews were analysed with no new themes arising from the interviews. This occurred after eight interviews and we completed a further two interviews to be confident that no new themes were emerging. We felt confident that we had enough rich data for a robust analysis.

### Data collection process

The interviews were completed by the lead author (RM), a physiotherapist currently working as an academic researcher towards completion of a PhD. The interviewer had no previous relationship with the participants and did not inform the participants of her background as a physiotherapist to avoid this influencing participant responses. Interviews were completed at a mutually agreeable time in the participants' own homes or in university meeting rooms where this was not possible. A topic guide was produced and followed throughout each interview to ensure consistency between interviews. Examples of the questions asked can be found in box 1. Field notes were taken throughout the data collection and analysis to maintain reflexivity during the project. Interviews were completed from 12 October 2018 to 03 April 2019 and continued until no new themes were emerging from the data.

### Data analysis

Interviews were recorded with the use of an encrypted digital audio recorder to which only the lead researcher had access. Interview recordings were then downloaded to a secure server and were password protected with access only to the lead researcher. The interviews were transcribed verbatim and all identifiable information was removed to ensure participant confidentiality. Interview transcripts were stored on a secure server and pseudonymised by a unique study ID number. Once transcribed, interviews were uploaded to NVivo (V.12, QSR International) for analysis. We used thematic analysis to analyse the data. A second researcher (ZHL) duplicated the coding process in a sample of four of the interviews to ensure dependability and consistency in the analysis process.[15] Transcripts were coded independently and then each interview transcript was discussed between the researchers to ensure agreement. Any sections of the transcripts which we did not agree on the coding for were discussed to reach consensus agreement on the most appropriate code to use in that section. We performed the analysis concurrently with the data collection and interviews were terminated when no new themes were emerging from the data.

### Patient and public involvement

Two patient and public involvement representatives from Warwick University User Teaching and Research Partnership were involved in the design and conduct of the randomised controlled trial, and provided consultation during protocol development. The representatives took an active role in reviewing and commenting on the interview study processes and associated burden of the study on participants. They provided consultation on the patient-facing materials used in this qualitative study, including the patient information sheet, invite letter and topic guide to ensure suitability. The representatives will also be collaboratively involved in planning the dissemination of results to participants alongside the results of the AIR Trial when available.

## RESULTS

A total of 19 participants was invited to take part in this study. Nine declined participation and 10 participants were recruited and interviewed as part of this study. The participant recruitment flow chart is found in figure 1 which shows the reasons for non-participation in the interview study. The age range was 21–75 years with a mean of 51.6 years. After eight interviews, no new themes emerged from the data and therefore we completed two more interviews to ensure no further themes arose from the interviews, as per the conditions outlined in the methods. We terminated the interviews at 10 participants as there were no further themes in the remaining two transcripts. Participant demographics and injury information are found in table 2.

During analysis, eight themes emerged from the data; mobility, loss of independence, healthcare, ankle symptoms, sleep disturbance and fatigue, family and social life, psychological effects, and activities of daily living.

### Mobility

Participants described their difficulty in walking 'normally' and getting around, usually describing this as frustrating or inconvenient. Many spoke about their reduced mobility in the context of their weight-bearing restrictions and walking aids, which were usually discussed inextricably. The frustration caused by walking aids and weight-bearing restrictions were especially evident in the older participants of the study, many of whom described this as the most difficult part of their fracture for them.

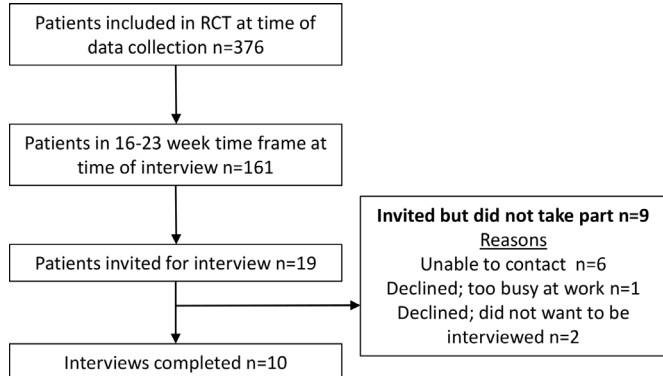

**Figure 1** Participant recruitment flow chart. RCT, randomised controlled trial.

Several individuals explained that the weight-bearing restrictions were too difficult to adhere to and described how they were not following the weight-bearing restrictions advised by their clinician for this reason. Individuals described using walking aids as slow and hard work, requiring frequent rests and some noted their frustration with the sudden inability to carry things. However, people also spoke of their walking aids as a necessary inconvenience, acknowledging that they were essential during periods of weight-bearing restrictions. Older participants discussed a fear of falling, usually when leaving the house or out in busy public places. Some described the difficulty in not being able to drive, explaining how that was an important factor in their recovery for them.

Pt09: *Just stuck here…I felt like a prisoner in my own home I think for the first 4 weeks…*

Pt07: *I used to dream of walking (the dog) down the park. Yeah…I would be…you know…dream about it. Erm (pause) just…just to be on two feet that was my sort of you know (pleading) "please let me get onto two feet"*

Pt03: *Yes because you know I thought crutches were easy things to use. But as I say to have to hop at my age is a very hard thing to do…*

Pt10: *You couldn't go more than a hundred…two hundred metres without stopping because it just puts so much pressure on your hands.*

Pt01: *the worst was when I wasn't able to drive; once I was able to drive again I think that was a turning point.*

### Loss of independence

A loss of independence and subsequent reliance on others was spoken about by all participants, which was required for household tasks, care of dependents or transport. For some this was a significant source of frustration. Some individuals identified the need to rely on others as a cause of low mood and described how the ability to regain independence was of vital importance for their mental well-being. For others, the need to rely on others was less bothersome and something they described as adjusting to. Some individuals described how the need to rely on others caused tensions within

the relationships. Despite needing to rely on others for some things, people also described adapting the way they did things in order to retain their independence as much as possible.

Pt01: *I think that's the thing that got me down the most having to rely on others to come and take you out and you know which…which people were absolutely brilliant but I'm quite independent and…and I…that was hard*

Pt09: *…for mental reasons it's good to get back to normal errm that was quite important for me to feel as though I was able to take charge of my own life again. Erm yeah rather than relying on other people.*

Pt10: *I dunno you just get a bit claustrophobic everyone doing everything for you.*

Pt05: *The worst part was not being able to do things when I wanted how I wanted that was the worst definitely.*

Pt05: *He was having to do everything and I would sit and stare at the washing up and I couldn't do it and it was so frustrating because I like a clean house.*

### Ankle symptoms

Individuals spoke of troublesome symptoms around their ankle to varying degrees, including pain, skin changes, wound issues, swelling, reduced movement, and loss of strength and muscle bulk. While many individuals felt it was important to not be in pain, many described their pain as manageable and controllable, which didn't prevent them from performing functionally. Skin changes including dry skin around the ankle were noted, particularly by those participants who received a plaster cast. One individual became very concerned with the development of pressure sores from the plaster cast which kept them awake at night. Several people discussed the swelling in their foot and ankle, often attributing this to a difficulty in finding suitable footwear. Almost all participants interviewed described a heightened awareness of their ankle, especially when discussing being out in public spaces, walking on uneven surfaces or returning to physical activity.

Pt06: *I still carried on. I mean really it was not…on a scale of one to ten…it was never more than a sort of four or a five to be honest.*

Pt01: *…and like I say of an evening if I…to come in from work not feel that it's swollen and not feel that it's uncomfortable erm then I think that perhaps that would be a hundred percent recovery.*

Pt05: *Err that…the first time I think I was really concerned about the heel I was convinced it was numb because the cast was too tight and the fact that I couldn't look at it…I couldn't get to it…I couldn't see it. Errm I was convinced absolutely convinced that I was getting all sorts of you know pressure sores…blisters.*

Pt09: *I've got very good flexibility in my ankle but not much strength. So that's what I'm working on.*

**Table 2** Participant demographics and injury information

| Participant ID | Time since injury | Participant age (years) | Gender (M/F) | Fracture management (operative or non-operative) | Allocated interventions | Any other injuries | Mechanism of injury | Occupation |
|---|---|---|---|---|---|---|---|---|
| 01 | 22 weeks | 48 | F | Operative | Functional brace | None | Low energy fall | Teacher |
| 02 | 22 weeks 6 days | 59 | M | Operative | Plaster cast | None | Low energy fall | Maintenance worker |
| 03 | 21 weeks | 75 | F | Non-operative | Plaster cast | None | Low energy fall | Retired, volunteers for charity |
| 04 | 19 weeks, 1 day | 37 | M | Non-operative | Plaster cast | None | Low energy fall | University lecturer |
| 05 | 21 weeks | 29 | F | Operative | Plaster cast | Contralateral ankle sprain | High energy fall doing gymnastics | Nurse |
| 06 | 20 weeks, 4 days | 60 | M | Non-operative | Functional brace | None | High energy fall while wakeboarding | Retired |
| 07 | 21 weeks, 6 days | 73 | F | Operative | Plaster cast | Ipsilateral calcaneal fracture | High energy fall from loft ladder | Part-time job in a shop |
| 08 | 21 weeks, 4 days | 45 | M | Non-operative | Functional brace | Contralateral ankle sprain | On a push bike which collided with a lorry | Works for car manufacturer |
| 09 | 22 weeks, 3 days | 69 | F | Operative | Functional brace | Wrist fracture | Low energy fall | Retired |
| 10 | 20 weeks, 2 days | 21 | M | Operative | Functional brace | None | High energy fall from moving vehicle | Undergraduate Student |

## Sleep disturbance and fatigue

Participants described disturbed sleep and increased fatigue in their recovery period from the ankle fracture. Individuals described difficulty getting to sleep or waking during the night due to pain. Those who had a plaster cast described a difficulty in getting comfortable at night because of this. Some described a general increase in fatigue because of the increased effort that walking took. Many spoke about sleep in the context of their medication, with some individuals using pain medication to aid with sleep. Others described how the sleep issues and subsequent tiredness affected their performance at work. Some felt that the effect on sleep was one of the more important factors in their ankle fracture recovery.

> Pt04: *Erm I think the loss of sleep was the worst. Yeah…it wasn't it wasn't even so much the pain itself as the fact that I wasn't sleeping properly and I was tired all the time for a few weeks I think that was the worst.*

> Pt02: *I think it stopped me from…yeah I think it stopped me from sleeping so much. I just couldn't get it comfortable at all… it's when it was in the cast was the main thing… getting it comfortable in the cast.*

## Psychological effects

Psychological effects were discussed to varying degrees between participants in this study. Many described their difficulty moving around and need to rely on others as causing feelings of frustration and this was a commonly described emotion. Some described feelings of depression and low mood attributed to their injury and the limitations it caused. One individual described an emotional lability during their recovery period, explaining how they would cry a lot more readily than usual throughout this time. Several people described an anxiety regarding the long-term function of their ankle. Younger participants particularly described this anxiety when discussing valued activities such as sports and leisure. The older participants were more concerned with getting back to usual function in terms of walking and driving, explaining their hopes of regaining their prefracture level of independence. There were some individuals who did not report any mood changes as a result of their fracture. Some individuals reported issues with body image and voiced concerns regarding their inability to exercise and the impact that might have on their weight. Others spoke of their injury and associated limitations in relation to feeling old or referencing the ageing process when discussing their recovery.

> Pt09: *Yeah. Tearful. Yeah. Not so much when I was out 'cos you don't do that. But yeah. It's just a horrible feeling… just…I don't…it's so difficult to be able to pin point exactly what it is that actually brings the tears on. I was low…very low.*

> Pt03: *Yeah but it did make me depressed at times yeah.*

> Pt05: *Any anxieties I had at that time were around long term recovery. I mean I probably could have been happier but I wouldn't have gone so far as to say I was actually depressed.*

> Pt08: *I was anxious…Yeah but it's just it…well it's alright now but at the time I remember thinking…I was…I was quite frightened 'cos it was really painful.*

> Pt05: *…and that is so important to me to be able to get back to that…to be fit you know so I'm quite weight conscious. I'm very conscious of the fact that I don't want to get fat sitting around and not doing anything.*

> Pt03: *Errm one of these elderly people's push about things in the house (pause) and I just get it into my mind that I'm not old enough for one of them yet (laughs) so that was very disheartening yeah to use that.*

## Activities of daily living

Participants described their difficulty or inability to complete their activities of daily living, such as personal care, household tasks, work and leisure activities. When discussing personal care and washing and dressing, many people spoke of finding new routines and adapting to new ways of doing things. Individuals discussed these in relation to their walking aids and weight-bearing status, stating that these factors meant that the process of washing and dressing took much longer. Those who received the functional brace spoke of the benefit of removing the brace for washing. Those who received a plaster cast discussed the need to use a cast cover for washing. Some female participants spoke of the frustration of not being able to shave their leg due to the irremovable cast.

> Pt09: *…fortunately they could take me to the toilet on a like a wheelie commode thing which sort of slots over the toilet so that wasn't too bad in that respect but it's still…(sighs) it's something we don't like I suppose isn't it? You know the personal things…going to the toilet…washing…somebody had to shower me but you know I…I was OK.*

> Pt04: *Getting in the shower was a pain…I'm kind of hopping over a slippy floor in the shower erm with a weird pose to make sure I'm not putting too much weight on my leg. Erm so it took much longer to shower.*

> Pt05: *I wanted to shave my leg…I wanted to wash it properly…the first thing I did when I got home I took it off had a shower shaved my leg it was disgusting. My other half nearly divorced me over the state of my leg! (laughs)*

Household tasks were severely restricted by participants' injuries and many spoke of these in the context of their walking aids, as they did not have hands free to carry things. Many spoke of relying on others for completion of essential tasks such as meal preparation and grocery shopping. Some spoke of adaption in doing things such as housework by getting on their hands and knees to complete tasks. Many people described how others took on the majority of the housework and caring for any dependents. Some people found a source of frustration in the standards of the other person completing the housework, which did not match their own personal standards.

Pt10: *I remember the most frustrating thing…it was just the little things just having a big bag of washing on the top floor and I couldn't put it on the handle of the crutch and peg it down so I had to call someone up and they had to come and cart it down for me.*

Pt05: *Yeah well I was lucky my wife basically took over everything. We try and share most things and I was…I was totally just not contributing at all so I mean as typical evening routine now I would help get the kids to bed I would help bath them I'd help give them dinner and while she's getting the baby to sleep I'd do the washing up and hoover and erm pack the bags for tomorrow things like that so for a few weeks that almost went out the window.*

Pt07: *Erm yes I discovered that if I got on my knees…hands and knees (pause) I could actually…not hoover…but I could get a stick brush and sweep the carpet. Erm I did that for quite a while. And then you had to make sure that you were near something so that I could get back up again. I also did gardening on my hands and knees because it was…to see it just going…it was heart-breaking. And I thought 'right if I could get out there I'm sure if I'd got something to kneel on I can actually do that' and I did.*

In terms of leisure time, many people were restricted from participating in their usual leisure activities due to their injury and used this free time for more sedentary activities instead. For some this was acceptable but this was a source of frustration for others, who described a dislike for 'sitting around'. Those who were normally physically active discussed their anxieties with returning to these activities and discussed feelings of caution and heightened awareness in their ankle associated with their return to sports and exercise. Those with more sedentary hobbies reported little or no impact on these. Several people spoke of having to miss holidays due to their injury. For those who worked, many people spoke of needing to reduce working hours or be off work due to their injury. Those who were able to work from home discussed doing so throughout the recovery period. Some discussed the financial implications of not working, which was a concern for some.

Pt06: *Errm errrm but apart from that it was obviously a massive limitation on doing all the sport I like to do. Errm so that was….it sort of drove me a bit mad.*

Pt09: *…you know when you've got the opportunity maybe to sit and watch television all day or read all day or whatever but you don't…mentally you don't feel like doing that either?*

Pt07: *I mean I do sewing…I make bags and that…well of course I just hadn't got the strength in that foot to press the pedal down on the machine (pause) so I couldn't do…yeah anything like that.*

## Family and social life

Many individuals discussed the impact that their ankle fracture had on their usual social and family life. The majority of individuals described how they were unable to go out independently to meet others and instead people would come to visit them. Some described a reduction in alcohol consumption because of this and additionally due to the pain medication they were taking. Some described how their social life improved as they saw family more as they were always checking in to help them. Some individuals described how their low mood meant they didn't feel up to socialising as much as usual. In terms of family life, many of the participants spoke of a need to adapt usual roles and responsibilities in light of their injury. A few individuals mentioned the tensions this could sometimes cause in their relationships with others. The impact on the family and family activities was also discussed, such as adapting childcare responsibilities and activities usually done as a family. Several individuals took time to explain their concern with the strain and pressure this put on other members of the household or wider family, who were taking on more workload than usual. Those with childcare responsibilities spoke of the psychological impact of not being able to perform the role they would usually perform for their children.

Pt09: *I was totally reliant either on (husband) or on friends to pick me up…take me out but again you don't feel like doing the things. Erm people would say "Ooh you know come and do this.'"And you think "Ohh I don't want to be out in company'"but we did I mean we forced ourselves to. Erm so yes it did impact on it but only probably because we allowed it to in the sense that we didn't want to go anywhere.*

Pt01: *I suppose the most frustrating thing for me really was that erm (name of child) my eldest was doing his A-Levels and so I was unable to take him backwards and forwards because, he was…he was at (school name), I was unable to drive him to his exams so my dad he was brilliant he stepped in but…just little things like that it just makes you feel a little bit (pause)…you….psychologically you feel like you're not able to perform the role that you normally perform sort of thing so…*

Pt 05: *Having to spend so much time with my mum that we started to grate on each other's nerves after a while and in the end you just think "I'd rather just stay indoors than have to go round and have another argument with mum because I've seen her every day for the last 2 weeks".*

## Healthcare

Individuals discussed their experiences of the healthcare they received for their ankle fracture. Many of the participants praised the services provided and the staff providing them during their recovery period. Those who required an inpatient stay for their injury described these experiences as lonely or difficult, usually due to other patients in the ward with them. Fracture clinic experiences were generally positive, with some individuals expressing confusion about seeing a different clinician to the one who performed their fracture fixation operation. Those who received physiotherapy generally recounted this as a useful experience, describing it as solution-focused and helpful in providing education and reassurance about their injury. Others described physiotherapy as slow and

hard work. Those who were not offered physiotherapy explained how they felt they would have benefited from it or felt they still needed some physiotherapy intervention. For one individual, when asked about the factors of most importance to them, they responded that the radiographic outcome was important and being able to see that the bone had healed on the radiographs was the most important factor to them in their recovery.

Pt02: *Well when they took the cast off the only thing that did concern me; I didn't have the same surgeon looking at it. I was expecting the same one as he did it just to say it was ok but erm I thought that was pretty strange. Usually you get the same surgeon all the way through the same doctor all the way through.*

Pt01: *She (physiotherapist) was absolutely brilliant really, really good and I think that just helped as well just having somebody you know showing you the things you could do but also say… you could just give you the confidence that yes you are fine to walk on it it's not going to (pause) do anything or whatever so that…I think…I think the physio is really important.*

Pt10: *No that's the only thing really they never offered me physio. Erm whether they felt I didn't need it or anything I'm not sure. Or whether that's something you have to pursue privately I don't know. But no they never really offered it or spoke about it… but my Mum said "you should probably get some" but I just never really followed it up.*

Pt08: *Oh seeing the radiographs 'cos I wanted to see the bone…solid again. And every time I kept going back and just seeing the gap…that was the target I wanted to see a radiograph that looked like it didn't have a split down the bone. But that was the initial thought it was like let's see the radiographs…when that crack's gone I'm…I'm well again*

## DISCUSSION

The results of this qualitative study show that individuals experience a wide range of concerns related to their ankle fracture, including the usual symptoms of pain and reduced mobility. Individuals also described effects on psychological well-being, sleep, their sense of independence, and family and social life. The factors of most importance discussed by the participants here ranged from regaining independence, improving sleep quality and quantity, ability to drive or get out and about, no longer requiring the use of walking aids or needing to follow weight-bearing restrictions. Some were also concerned with the radiographic outcome during their recovery. There were variations in experience in relation to the age of participants; older individuals described a more intense difficulty in adhering to weight-bearing restrictions, coping with the loss of independence and reported more severe psychological effects than the younger individuals interviewed here.

Considering the significant difficulty that older individuals face in tolerating weight-bearing restrictions there is a need for further research into the most appropriate weight-bearing protocols for ankle fractures, which is unclear and

there is evidence of inconsistency in clinical practice.[16–18] In some cases, weight-bearing may provide more benefits than harm and further evaluation of these protocols is warranted. In instances where weight-bearing restrictions are deemed essential, consideration of alternative or innovative walking aids would be helpful to ensure restrictions are able to be adhered to. The provision of physiotherapy between participants was inconsistent and the evidence for rehabilitation protocols following ankle fracture are similarly unclear in the literature.[16] Research to identify the most effective physiotherapy interventions, for which patients and during what time frame, is warranted to standardise care. The prevalence of psychological effects such as anxiety and depression reported in this study indicates that a more holistic approach to intervention is required following trauma. Furthermore, the importance of returning to driving is key for some patients and there is often a lack of definitive guidance given to patients regarding this.[19]

This study compares to another qualitative study completed with those with ankle fracture[10] who also demonstrated wide-ranging effects on individuals with ankle fracture, including social and psychological impacts as well as activities of daily living. There were some differences of this study, in that here we focused only on patients with an ankle fracture, whereas the previous research also interviewed clinicians. This article adds the concepts and factors most important to individuals with an ankle fracture, contributing further to this research area. Other research into patient experience of hip fracture showed similar thematic results including mobility and psychological effects.[20] A similar study has also been completed exploring the patient experience of ankle reconstruction for ankle osteoarthritis[21] who discussed a central theme of vigilance of their affected ankle, which agrees with findings here in relation to individuals feeling aware or cautious of their ankle injury during recovery. Another article focussing on older women with vertebral fracture in Sweden also compares to results found here, showing the importance for individuals to maintain their independence as much as possible when recovering from a fracture.[22]

The strengths of this study include the exploration of a burdensome condition which is increasing in prevalence. Exploring the factors of most importance to individuals with ankle fracture in such a widely studied injury is important in ensuring we are collecting relevant and important outcomes for individuals with this injury, as well as providing clinical care which is sensitive to issues most pertinent to them. We used a purposive sample to interview a representative sample of the population of adults with ankle fractures and duplicated the data analysis for purposes of consistency. A weakness of this study is that the timing of the interviews were limited by the primary outcome time point of the trial. If this was not a constraint, it would have been beneficial to interview at regular time periods from time of injury to ensure that participants could be interviewed throughout the recovery period, rather than requiring the participants to recall information from their whole recovery experience. Furthermore, there were nine individuals who were invited but did not take part, three of whom declined to be interviewed. There

is a possibility that these individuals might differ significantly from those who agreed to participate, for example, have had more difficulty throughout their recovery than others. Finally, the lead researcher (RM) and second coder (ZHL) are both physiotherapists by background. While every effort was made to ensure researcher reflexivity and reduce bias throughout the process of the study, as with all qualitative enquiry, the researchers' professional backgrounds and personal experiences will likely have introduced bias throughout the collection, analysis and interpretation of data.

## CONCLUSION

Results presented here enable a greater understanding of the lived experiences of individuals with this injury to allow for clinicians to better plan and implement appropriate patient-centred management strategies. This research shows that individuals with ankle fracture experience issues not only with mobility and pain, but also with adhering to weight-bearing restrictions, psychological effects and a profound impact on their sense of independence. Further research should focus on the most effective weight-bearing and rehabilitation protocols for this patient population, which can vary in clinical practice. Furthermore, the results here suggest that older patients may experience these effects more severely than younger individuals. The results of this study will enable those involved in clinical research for interventions for ankle fracture to select the most appropriate patient-centred outcome measures which assess items most important to patients.

**Acknowledgements** The authors thank the participants of this study for giving up their time to speak to them about their experiences with their ankle fractures. The authors also thank the Patient and Public Involvement representatives, Karen Keates and Richard Grant for their contribution to this study. The authors also thank the AIR Trial Management Group for their guidance throughout this study.

**Contributors** RM, RSK and DRE developed the protocol. RM completed the participant screening, recruitment, data collection and interview transcription. RM and ZHL completed the data analysis. RM, DRE, ZHL and RSK contributed to the final manuscript. All authors reviewed and approved the final manuscript.

**Funding** RM is funded by a National Institute for Health Research (NIHR) Career Development Fellowship for this research project. This publication presents independent research.

**Disclaimer** The view expressed are those of the author(s) and not necessarily those of the NHS, the NIHR or the Department of Health and Social Care.

**Competing interests** RSK is a NIHR Senior Fellow, current member of the UK NIHR HTA CET board and NIHR ICA Doctoral panel and former member of the NIHR RfPB board.

**Patient consent for publication** Not required.

**Ethics approval** Ethical approval was gained through substantial amendment to the Ankle Injury Rehabilitation (AIR) Trial protocol (ISRCTN15537280) from the West Midlands Edgbaston NHS Research Ethics Committee on 07 September 2018 (reference: 17/WM/0239).

**Provenance and peer review** Not commissioned; externally peer reviewed.

**Data availability statement** No data are available.

**ORCID iDs**
Rebecca McKeown http://orcid.org/0000-0002-3502-2328
Rebecca Samantha Kearney http://orcid.org/0000-0002-8010-164X
David R Ellard http://orcid.org/0000-0002-2992-048X

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
