## [Reviewer comments · BMJ Open]

ARTICLE DETAILS

TITLE (PROVISIONAL)	What are the patient experiences of an ankle fracture and the factors of most importance in their recovery? A qualitative interview study.
AUTHORS	McKeown, Rebecca; Kearney, Rebecca; Liew, Zi Heng; Ellard, David

VERSION 1 - REVIEW

REVIEWER	Richard Buckley University of Calgary, Canada
REVIEW RETURNED	26-Aug-2019

GENERAL COMMENTS	This reviewer does not believe that saturation has been reached with the number of patients interviewed. What about minority populations, married, single, immigrants, laborers, office workers, athletes? This population of interviewees needs a much more careful description such that this study could be repeated or done in another population. What about driving a car? This seemed to be low on the list but in my experience in my extensive practice is one of the toughest problems. Is this not so in Britain? The names of the researchers are used and usually only the initials are used. A few references are missing as the references seem quite colloquial and not worldly. What about these references? 1) Dehghan N, McKee MD, Jenkinson RJ, Schemitsch EH, Stas V, Nauth A, Hall JA, Stephen DJ, Kreder HJ. Early weightbearing and range of motion versus non-weightbearing and immobilization after open reduction and internal fixation of unstable ankle fractures: a randomized controlled trial. Journal of orthopaedic trauma. 2016 Jul 1;30(7):345-52. 2) Dehghan N, Mitchell SM, Schemitsch EH. Rehabilitation after plate fixation of upper and lower extremity fractures. Injury. 2018 Jun 1;49:S72-7. The Discussion seems short of real recommendations. There is no ordering of "most to least" important (this reviewer knows that it is a qualitative study but this can still be attempted to provide some context) factors. Who is most affected? Which demographics (assuming that you have a "saturated profile" which this reviewer doubts)?
--

	What about providing some solutions to the problems when injured? This could be answers to the problems of 1) full weight-bearing as most recent references state, 2) better walking aids like "scooters", 3) better casts like removable ones or splints rather than plaster casts, 4) more physio, 5) more education. All of these should be offered as at least a reflection of the authors thoughts from this qualitative research (just not leaving it up to the reader to guess what might be the answers).
--	---

REVIEWER	Jane Clemensen University of Southern Denmark
REVIEW RETURNED	21-Oct-2019

GENERAL COMMENTS	Love the paper. Its relevant courageous and well written, it takes the reader by the hand and lead you simple but elegantly through the study and the presentation of it. The discussion covers the many questions that arises when reading the results.\r \r Would be interesting if the authors repeated the study but with focus on patients on the other end - the non-privileged and vulnerable - where are they and then again a study doing the same but with the doctors perspectives. Waiting for those papers:-)\r \r Miss a conclusion.\r Few spelling problems.
---

REVIEWER	Alastair Younger, MB ChB MSc ChM FRCSC Department of Orthopaedics, Division of Distal Extremities, University of British Columbia Vancouver Canada Consultant Conflict statement from the American Academy of Orthopaedic Surgeons Alastair S E Younger, MD (Canada) Submitted on: 04/29/2019 Acumed, LLC: Paid consultant; Paid presenter or speaker; Research support American Orthopaedic Foot and Ankle Society: Board or committee member Amniox: Research support Bioventus: Paid consultant; Research support Canadian Orthopaedic Association: Board or committee member Cartiva: Research support CONMED Linvatec: Paid presenter or speaker Ferring Pharmaceuticals: Paid consultant; Paid presenter or speaker; Research support Foot and Ankle International: Editorial or governing board Stryker: Paid presenter or speaker Synthes: Research support Wolters Kluwer Health - Lippincott Williams & Wilkins: Editorial or governing board Wright Medical Technology, Inc.: Paid consultant; Paid presenter or speaker; Research support Zimmer: Paid consultant; Paid presenter or speaker; Research support
REVIEW RETURNED	26-Oct-2019

GENERAL COMMENTS

Thank you for your paper as it captures many aspects of ankle fractures that as physicians and surgeons we hear in clinic that do not get recorded in standard outcomes papers on ankle fracture treatment. Capturing these elements studied is important as we have to date failed to convey the disability and difficulty of recovering from foot and ankle injury such that resource is not dedicated towards it. This paper when published and similar using similar research methodology will go a long way towards advocating for this patient population.

With respect to the author make up it is a shame that you do not have a Orthopaedic surgeon with you as many of the comments I will make would likely have been solved had you had a surgeon as a co author. If this is possible I would suggest it as it would assist in making the manuscript more relevant to the treating environment.

abstract

Participant recruitment not clear - how do the authors know this is representative and generalizable with a sampling size of 10 patients?

Strengths and limitations: limitation - sample size and lack of statistical analysis

Background

Line 37 - Ankle fracture is not a fragility fracture in the young, and in the elderly is not considered a fragility fracture - therefore I would avoid reference to fragility fractures- the incidence and impact speaks for itself.

Introduction - there is much work done previously on outcomes of ankle fractures and the authors themselves are involved in this. A more detailed review of some of these outcomes would be helpful (what the anticipated outcome is, adverse event rate) and the limitations of the outcome recording from standard outcomes research, and why it fails to capture the patient experience - i.e. what makes this contribution different, why we should use this research methodology, and what is unique in this paper compared to others on the outcome of ankle arthritis treatment.

Sample size and patient selection

This is where I have the main concern over this paper: It is not clear to me who was invited to participate, and how the authors controlled for participation bias. Therefore are the results representative of their study population as a whole, and as a reader of the article would it be generalizable for my patient population: this journal is an international journal so this aspect is particularly important.

Can the authors please better explain who was invited to participate, how the invitations were made, how many declined, and was there a statistical significance in those that participated and declined. Can they show that this group is representative of the entire cohort they were selected from with regards to age, sex, surgical vs non surgical treatment etc. While the study is purposive if I were to repeat it I would have no idea who to approach, how to approach them, how to control for selection bias, how to ensure that the study population was representative. Further the entire cohort for the RCT needs to be better described to determine if the results are generalizable. So if I was to do a similar study here at my University what patient population would I select and how would I recruit them? A list or table of inclusion and exclusion criteria would be very helpful as used in the STROBE checklist. My concern is that the group selected for the study were not

representative of the population at large with ankle fractures and that those interviewed may be a unique subset because they were willing to be interviewed. For example if the entire population were half English speakers and half non English speakers and the interviews conducted in English then the study would only be relevant for English speakers which creates a Bias. This may be OK but it has to be outlined in the inclusion and exclusion criteria such that if the reader were treating a non English speaking population they could decide if the result was generalizable to their population.

Recruitment and consent - how many invited did not participate? what was the demographics of the non participation group? Why 10 people? Why not 5 or 50? what was the rationale for the sample size?

Method and direction of questions

The main concern of standard outcome reporting is that the issues that concern patients are not reported - and the need to do studies like this. Perhaps quote Pinsky who did a similar study on outcomes on ankle arthritis and identified areas in outcomes scales of relevance to patients (Swelling, sports). However the example questions while open ended still direct the patient and hence has a similar limitation to standard outcome questionnaires. Did the authors ask an open ended question such as "Is there any other effect of your fracture that we have not discussed" to try and understand what might be relevant that is purely patient directed not outcome scale or interview directed.

Page 7 of 26 - what do the authors mean by data saturation was reached after 8 interviews - that no more data may be recorded or that sufficient data to be relevant to the study methodology?

For data analysis (line 36) two researchers reviewed the data - however was there any test for validity or reliability of the data capture?

Results - clear - however I don't think that the quotes help and often detract from the scientific content and is more akin to journalism than scientific writing. I would suggest limiting to 1 quote per field.

Discussion - I as a surgeon looking after this patient population am constantly frustrated by the lack of resource dedicated to their recovery. This paper's strength is in potentially improving this care by better education and facility which ultimately may be more cost effective and lead to better outcomes. This is where a surgeon co author would be very helpful. The comment on fracture healing and seeing the bone unite indicates that we have to change our education. the comment on the variability of the surgeon in clinic also give insight as scheduling may not respect this. This paper may also assist in discharge planning, and further assist in preventing admission by better understanding the barriers to being at home if being treated as an outpatient. The concern of the elderly patient about falling is also an issue. Many of these concerns may in themselves be topics of follow up research and an outline of where to go (qualitative vs quantitative research) would be helpful. Further the limitations of cast treatment are better outlined and for some elderly patients they may have been better managed by percutaneous treatment (see White) as the RCT's on this topic (plate vs nail or non op vs op) fail to capture many of the concerns raised by patients in this study. Personally I have therefore questioned the RCT's and their validity and this paper further raises the question as to the policies determined by RCT's may be the wrong one because we are not asking the right

	questions. The primary outcome of Achilles tendon RCT's has been re rupture, whereas patients are likely not concerned about this but may be much more concerned about being able to go up and down stairs, stand up on tip toes or be able to pass the fitness test to be within the police or fire service. Knowledge of this may direct surgical innovation (less invasive techniques to restore mobility) or change the treatment recommendations based on age group. Hence we don't operate on something we may need to operate on because the study never understood the patient perspective. references add Pinsker doi: 10.1002/acr.24039. Pinskerdoi: 10.1177/1071100712460365. and white doi: 10.1302/0301-620X.98B9.35837. plus a more detailed review of quantitative outcomes of ankle fracture treatment
--	--

VERSION 1 – AUTHOR RESPONSE

Reviewer Comments

Reviewer 1:

Comment: This reviewer does not believe that saturation has been reached with the number of patients interviewed. What about minority populations, married, single, immigrants, laborers, office workers, athletes? This population of interviewees needs a much more careful description such that this study could be repeated or done in another population.

Response: Many thanks for your comment regarding our population included in this study. We have amended our use of the term saturation as this is misleading. We meant that no further or new themes emerged from the interviews after 8 patient interviews and used this as a guide for terminating recruitment. We have now made this clearer in our methods and results sections. We agree that there are many factors which could affect patient experience of this particular injury. We have utilised purposive sampling to account for as many factors as possible and have outlined these in detail in table 3 in the results section, which includes age, gender, fracture management, mechanism of injury, further injuries and occupation of individuals. We appreciate that we have not factored for all aspects which could affect experience in these individuals, however we feel that our approach is reasonable considering the specific case of sampling from an ongoing clinical trial in an UK setting.

Comment: What about driving a car? This seemed to be low on the list but in my experience in my extensive practice is one of the toughest problems. Is this not so in Britain?

Response: Many thanks for your comment regarding driving. We have included this as a factor of high importance to some patients in this study and this is outlined in the theme mobility. Whilst several people explained this as being an important factor to them in their recovery, not everyone did and it was not the most frequently discussed factor by the participants. This could be explained by the fact that not all participants interviewed used a car regularly or at all. For those which did, it was an important issue for them, and therefore we have included a quotation to support this finding in the relevant theme heading.

Comment: The names of the researchers are used and usually only the initials are used.

Response: Many thanks for your comments regarding the names of the authors. We have amended this to use initials only and we hope this improves the quality of the manuscript.

Comment: A few references are missing as the references seem quite colloquial and not worldly. What about these references?

1) Dehghan N, McKee MD, Jenkinson RJ, Schemitsch EH, Stas V, Nauth A, Hall JA, Stephen DJ, Kreder HJ. Early weightbearing and range of motion versus non-weightbearing and immobilization after open reduction and internal fixation of unstable ankle fractures: a randomized controlled trial. *Journal of orthopaedic trauma*. 2016 Jul 1;30(7):345-52.

2) Dehghan N, Mitchell SM, Schemitsch EH. Rehabilitation after plate fixation of upper and lower extremity fractures. *Injury*. 2018 Jun 1;49:S72-7.

Response: Many thanks for your suggestions for references to include in our manuscript. We agree that this information would be beneficial in supporting our arguments. I have included the first reference in the second paragraph of the background section and the second reference in the fifth paragraph of the discussion. We hope this improves the variety of our referencing for this article.

Comment: The Discussion seems short of real recommendations. There is no ordering of "most to least" important (this reviewer knows that it is a qualitative study but this can still be attempted to provide some context) factors. Who is most affected? Which demographics (assuming that you have a "saturated profile" which this reviewer doubts)?

Response: Many thanks for your comment regarding our discussion section. We have re-written our discussion section and hope that this makes our conclusions more appropriate for readers.

Comment: What about providing some solutions to the problems when injured? This could be answers to the problems of 1) full weight-bearing as most recent references state, 2) better walking aids like "scooters", 3) better casts like removable ones or splints rather than plaster casts, 4) more physio, 5) more education. All of these should be offered as at least a reflection of the authors thoughts from this qualitative research (just not leaving it up to the reader to guess what might be the answers).

Response: Many thanks for your comments regarding solutions to be presented in the discussion. We have rewritten our discussion to include some points which you suggest and hope this improves the overall quality of our manuscript and subsequent recommendations.

Reviewer 2:

Comment: Love the paper. Its relevant courageous and well written, it takes the reader by the hand and lead you simple but elegantly through the study and the presentation of it. The discussion covers the many questions that arises when reading the results.

Response: Many thanks for your comment on our study and manuscript and we are pleased that you enjoyed the content and presentation of our research article.

Comment: Would be interesting if the authors repeated the study but with focus on patients on the other end - the non-privileged and vulnerable - where are they and then again a study doing the same but with the doctors perspectives. Waiting for those papers:-)

Response: We thank you for your important comment on the sampling of patients included in this study and agree that it's of paramount importance that health research includes a wholly diverse patient population. Our purposive sampling has ensured that we are considerate for factors of age and gender to ensure we can capture a diverse range of individuals. Considering this is a patient experience question, we feel only asking the individuals with the injury, rather than including clinicians as well, is justified here.

Comment: Miss a conclusion.

Response: Many thanks for your comment on our manuscript regarding the conclusion. We have included our conclusion within the discussion section. We have re-worded our discussion section in some areas and hope this improves the clarity of the conclusion of the manuscript.

Comment: Few spelling problems.

Response: Many thanks for highlighting some spelling issues in the article. We have reviewed the article thoroughly and amended these as appropriate. We hope that this has improved the clarity of our research article.

Reviewer 3:

Comment: Thank you for your paper as it captures many aspects of ankle fractures that as physicians and surgeons we hear in clinic that do not get recorded in standard outcomes papers on ankle fracture treatment. Capturing these elements studied is important as we have to date failed to convey the disability and difficulty of recovering from foot and ankle injury such that resource is not dedicated towards it. This paper when published and similar using similar research methodology will go a long way towards advocating for this patient population.

Response: Many thanks for your insightful comment on our manuscript and the clinical relevance of its content. We agree and hope that qualitative research methods will continue to be employed to explore the patient experience of life-impacting injuries such as ankle fractures.

Comment: With respect to the author make up it is a shame that you do not have a Orthopaedic surgeon with you as many of the comments I will make would likely have been solved had you had a surgeon as a co author. If this is possible I would suggest it as it would assist in making the manuscript more relevant to the treating environment.

Response: Many thanks for your insightful comment regarding the author team. The team comprises of three physiotherapists and one academic methodologist. Whilst we agree that there could be more diversity in the professional groups of the authors, we feel that having three clinicians on the team is a strength in ensuring clinical relevance of the article. Furthermore, we would like to bring your attention to the acknowledgements section, which thanks the AIR trial Management group for their contribution to this study. This group includes a professor of orthopaedic surgery, a professor of podiatry and a professor of primary care/general practice with a special interest in musculoskeletal disorders. We therefore feel that we have the sufficient professional diversity within the contributors to ensure clinical relevance of this article.

Comment: abstract

Participant recruitment not clear - how do the authors know this is representative and generalizable with a sampling size of 10 patients?

Response: The aim of qualitative research is to gather detailed, rich information regarding the topic of interest and therefore purposive sampling is preferable to large, probabilistic samples (1). We would therefore not expect the sample to be completely generalizable to the population of interest. The purposive sampling strategy allows us to accommodate for factors which may affect patient experience, such as age, gender and the treatment provided for the fracture. It is not possible for us to purposively sample for all factors which may affect outcome, therefore we feel the approach to the factors we can purposively sample for have been taken into account. The sample size was based upon the point at which no new themes emerged from the data, which was after eight interviews.

Comment: Strengths and limitations: limitation - sample size and lack of statistical analysis

Response: Many thanks for your comment on the sample size and lack of statistical analysis in our article. As we used qualitative methodology in this study, the formulation of a sample size through statistical methods is not appropriate for this nonprobabilistic sample. Furthermore, statistical analysis does not apply to explorative qualitative research which we present here, therefore we do not consider this a limitation of this research. We took guidance from multiple methodological sources within the qualitative literature (2-4) for reference.

Comment: Background

Line 37 - Ankle fracture is not a fragility fracture in the young, and in the elderly is not considered a fragility fracture - therefore I would avoid reference to fragility fractures- the incidence and impact speaks for itself.

Response: Many thanks for your comment on our use of the term fragility fractures. There is debate in the literature as to whether ankle fractures can be considered as a fragility fracture. We have therefore removed some mentions of this and we hope this improves the background section of our manuscript.

Comment: Introduction - there is much work done previously on outcomes of ankle fractures and the authors themselves are involved in this. A more detailed review of some of these outcomes would be helpful (what the anticipated outcome is, adverse event rate) and the limitations of the outcome recording from standard outcomes research, and why it fails to capture the patient experience - i.e. what makes this contribution different, why we should use this research methodology, and what is unique in this paper compared to others on the outcome of ankle arthritis treatment.

Response: Many thanks for your comment on the background section of the article. We have chosen not to include any information of the anticipated outcome or adverse event rate due to this being outside of the scope of this article. We have included some more information on the rationale of the study and we hope this improves the background section of our manuscript.

Comment: Sample size and patient selection

This is where I have the main concern over this paper: It is not clear to me who was invited to participate, and how the authors controlled for participation bias. Therefore are the results representative of their study population as a whole, and as a reader of the article would it be generalizable for my patient population: this journal is an international journal so this aspect is particularly important.

Can the authors please better explain who was invited to participate, how the invitations were made, how many declined, and was there a statistical significance in those that participated and declined. Can they show that this group is representative of the entire cohort they were selected from with

regards to age, sex, surgical vs non surgical treatment etc. While the study is purposive if I were to repeat it I would have no idea who to approach, how to approach them, how to control for selection bias, how to ensure that the study population was representative. Further the entire cohort for the RCT needs to be better described to determine if the results are generalizable. So if I was to do a similar study here at my University what patient population would I select and how would I recruit them? A list or table of inclusion and exclusion criteria would be very helpful as used in the STROBE checklist. My concern is that the group selected for the study were not representative of the population at large with ankle fractures and that those interviewed may be a unique subset because they were willing to be interviewed. For example if the entire population were half English speakers and half non English speakers and the interviews conducted in English then the study would only be relevant for English speakers which creates a Bias. This may be OK but it has to be outlined in the inclusion and exclusion criteria such that if the reader were treating a non English speaking population they could decide if the result was generalizable to their population.

Response: Thank you for your important comment on our patient selection throughout this research project. We agree that we did not make the inclusion and exclusion criteria of the randomised controlled trial clear enough in the article and therefore have added this to the participant identification section of the manuscript. With regards to the identification and invitation process, this is detailed in the patient recruitment flowchart in Figure 1. We realise we have not signposted the reader to this figure in the main text so we have added this along with some explanatory text at the beginning of the Results section. We did not look at a statistical significance between those who participated and those who declined as this is a nonprobabilistic sample and therefore not appropriate to complete here. Because we used a purposive sample, we are able to demonstrate that we used a wide range of individuals with differing experiences. This is demonstrated in table 3, which shows that there is a sufficient spread of ages (21-75years), genders (5 male, 5 female) and fracture management (6 operative, 4 non-operative, 5 plaster cast, 5 functional brace). The purposive sample does not aim to create a sample that is generalizable to the population, but rather aims to provide a rich explorative evaluation of the topic in question. Whilst we have made every effort to employ purposive sampling for a range of factors, understandably we are not able to sample for every factor which could affect outcome such as ethnic background, languages spoken and socio-economic status. However, we feel we have accounted for as many factors as reasonably possible. I hope this makes the patient selection for this project clearer.

Comment: Recruitment and consent - how many invited did not participate? what was the demographics of the non participation group?

Response: Many thanks for your comment regarding the individuals who were invited but did not participate in the study. As mentioned in response to a previous comment, I have included a signposting statement in the results section to the patient recruitment flowchart (figure 1) which details the number of people who did not participate and the reasons for this. We have chosen not to include the demographics of those who did not participate as the individuals have not given their consent for their data to be used in this way for this study.

Comment: Why 10 people? Why not 5 or 50? what was the rationale for the sample size?

Response: Many thanks for your comment on the sample size used in the study here. The sample size here is based upon the point at which no new themes emerged from the data, which is frequently described in the qualitative literature as the criteria for terminating data collection (2). We reached this point at eight interviews and completed two more to ensure that no new themes emerged from these interviews. I have added some text in the results section to explain that the emergence of no new themes was the criteria at which data collection was terminated.

Comment: Method and direction of questions

The main concern of standard outcome reporting is that the issues that concern patients are not reported - and the need to do studies like this. Perhaps quote Pinski who did a similar study on outcomes on ankle arthritis and identified areas in outcomes scales of relevance to patients (Swelling, sports). However the example questions while open ended still direct the patient and hence has a similar limitation to standard outcome questionnaires. Did the authors ask an open ended question such as "Is there any other effect of your fracture that we have not discussed" to try and understand what might be relevant that is purely patient directed not outcome scale or interview directed.

Response: Many thanks for your comment on the method and direction of questions asked to participants in this study. We did ensure to ask participants open ended questions as well as questions which had more of a direction. We feel that the breadth and depth of patient experiences we have reported here over the eight themes demonstrate the appropriateness of questioning, with many factors such as sleep issues, psychological effects and loss of independence not appearing on the ankle fracture specific outcome questionnaires which are used routinely in clinical trials for this injury. We hope this has clarified this aspect of our study.

Comment: Page 7 of 26 - what do the authors mean by data saturation was reached after 8 interviews - that no more data may be recorded or that sufficient data to be relevant to the study methodology?

Response: Many thanks for your comment on our results section. To clarify, we mean that after 8 interviews, no new themes emerged from the data – we have removed the term data saturation as it is misleading. We have rewritten parts of this section and hope this clarifies the terminology we use here.

Comment: For data analysis (line 36) two researchers reviewed the data - however was there any test for validity or reliability of the data capture?

Response: Thanks for your comment regarding the data analysis. As we mentioned, we performed duplicate coding by a separate researcher in a sample of interview transcripts included in this study. We did not perform any test on reliability of coding between researchers. We have included a more detailed description of the coding process and hope this makes our methods section clearer.

Comment: Results - clear - however I don't think that the quotes help and often detract from the scientific content and is more akin to journalism than scientific writing. I would suggest limiting to 1 quote per field.

Response: Thanks for your comment on the quotations used in the manuscript. Standard 17 of the Standards for Reporting Qualitative Research, which we used to report this research, states that results/findings should be "linked to Empirical Data using evidence (e.g. quotes, field notes, text excerpts) to substantiate analytic findings" (5). Therefore we feel that the quotations are necessary to fulfil this criteria of the reporting checklist and have decided not to remove any quotations for this reason.

Comment: Discussion - I as a surgeon looking after this patient population am constantly frustrated by the lack of resource dedicated to their recovery. This paper's strength is in potentially improving this care by better education and facility which ultimately may be more cost effective and lead to better outcomes. This is where a surgeon co author would be very helpful. The comment on fracture healing and seeing the bone unite indicates that we have to change our education. the comment on the variability of the surgeon in clinic also give insight as scheduling may not respect this. This paper may also assist in discharge planning, and further assist in preventing admission by better understanding the barriers to being at home if being treated as an outpatient. The concern of the elderly patient about falling is also an issue. Many of these concerns may in themselves be topics of follow up research and an outline of where to go (qualitative vs quantitative research) would be helpful. Further the limitations of cast treatment are better outlined and for some elderly patients they may have been

better managed by percutaneous treatment (see White) as the RCT's on this topic (plate vs nail or non op vs op) fail to capture many of the concerns raised by patients in this study. Personally I have therefore questioned the RCT's and their validity and this paper further raises the question as to the policies determined by RCT's may be the wrong one because we are not asking the right questions. The primary outcome of Achilles tendon RCT's has been re rupture, whereas patients are likely not concerned about this but may be much more concerned about being able to go up and down stairs, stand up on tip toes or be able to pass the fitness test to be within the police or fire service. Knowledge of this may direct surgical innovation (less invasive techniques to restore mobility) or change the treatment recommendations based on age group. Hence we don't operate on something we may need to operate on because the study never understood the patient perspective.

Response: Many thanks for your comment on our manuscript and its strengths. Please see our previous comment with regard to the professional diversity of the authors and the research contributors. We have re-written our discussion section and hope this improves the clarity of the the conclusions and recommendations of this article.

Comment: references

add Pinsker doi: 10.1002/acr.24039.

Pinskerdoi: 10.1177/1071100712460365.

and white doi: 10.1302/0301-620X.98B9.35837.

Response: Many thanks for your suggestion of these articles for inclusion in our manuscript. We have included a reference to one of these in the third paragraph of the discussion section, describing the similarities of the author's findings of this article with ours. We thank you for drawing our attention to this article.

Comment: plus a more detailed review of quantitative outcomes of ankle fracture treatment

Response: Many thanks for your comment on our article and suggestion of adding a more detailed review of quantitative outcomes of ankle fracture treatment. We have decided not to include this because it is outside the scope of this research article. There are many components of ankle fracture intervention and rehabilitation and a review of the outcomes of these would not be possible in the word count we have available.

References

1. Tuckett AG. Qualitative research sampling: the very real complexities. *Nurse Res.* 2004;12(1):47-61.
2. Guest G, Bunce A, Johnson L. How Many Interviews Are Enough?: An Experiment with Data Saturation and Variability. *Field Methods.* 2006;18(1):59-82.
3. Braun V, Clarke V. *Successful Qualitative Research. A Proactical Guide for Beginners.* . London: Sage Publications; 2013.
4. Marshall MN. Sampling for qualitative research. *Family Practice.* 1996;13(6):522-6.
5. O'Brien BC, Harris IB, Beckman TJ, Reed DA, Cook DA. Standards for Reporting Qualitative Research: A Synthesis of Recommendations. 2014;89(9):1245-51.

VERSION 2 – REVIEW

REVIEWER	Dr. Richard Buckley University of Calgary, Canada Foothills Medical Center
REVIEW RETURNED	10-Nov-2019

GENERAL COMMENTS	Methods - it is not at all clear that operative ankles are in the study. Please make this clear in inclusion criteria. - Patient and public involvement - names should not be used here- acknowledgement at the end only. Results - loss of independence - 5th sentence - needs rewriting. References - need a good reference to return to driving eg. Archives of Orthopaedic and Trauma Surgery (2018) 138:1691–1697 https://doi.org/10.1007/s00402-018-3037-3 TRAUMA SURGERY Return to car driving is safe 6 weeks after operative treatment of right ankle fractures Sean W. L. Ho¹, Michael Yam¹, M. L. Chan², Ernest B. K. Kwok¹ Discussion - put strengths and limitations of the study at the end of the discussion. It is a bit jumbled now as it is. Rewrite this section as the end of the discussion does not fit there. Conclusion - add a title - Conclusion - and make a strong conclusion here. It is wimpy now. Make a strong conclusion like the abstract states eg. driving problems, loss of independence and problems of the elderly.
--

REVIEWER	Jane Clemensen University of Southern Denmark
REVIEW RETURNED	13-Nov-2019

GENERAL COMMENTS	Reflections and corrections are satisfied
---

REVIEWER	Alastair Younger University of British Columbia Canada Conflict statement Alastair S E Younger, MD (Canada) Submitted on: 04/29/2019 Acumed, LLC: Paid consultant; Paid presenter or speaker; Research support American Orthopaedic Foot and Ankle Society: Board or committee member Amniox: Research support Bioventus: Paid consultant; Research support Canadian Orthopaedic Association: Board or committee member Cartiva: Research support CONMED Linvatec: Paid presenter or speaker
-----------------	---

	Ferring Pharmaceuticals: Paid consultant; Paid presenter or speaker; Research support Foot and Ankle International: Editorial or governing board Stryker: Paid presenter or speaker Synthes: Research support Wolters Kluwer Health - Lippincott Williams & Wilkins: Editorial or governing board Wright Medical Technology, Inc.: Paid consultant; Paid presenter or speaker; Research support Zimmer: Paid consultant; Paid presenter or speaker; Research support
REVIEW RETURNED	30-Nov-2019

GENERAL COMMENTS	First - I really like the study and its message. However there is much work to be done to bring it to an appropriate standard for a journal. In particular there is no way that this study could be repeated based on the information presented. Specifically there is no way that I could repeat the study in my institution based on the information presented. Do I just walk down the hall, find 10 people with ankle fractures and bring them into a room at an assigned time and start asking questions? I need to know how to determine my sample size. If it is because I have enough budget to only interview 10 people that is fair enough - that is then the rationale for a sample size of 10. However there has to be a reason it is 10, not 1 or 100. Or is it based on the number of fractures seen in a region - and then determined that 10 is appropriate to be representative of that population. Once the number of participants has been determined (in studies I usually do we determine this via a power analysis to ensure that we have enough patients to know that the conclusions are valid) the selection of participants has to be clearly outlined so that bias is controlled. Review of the demographics of the 10 patients would indicate that they are representative but a table showing that the study group is at least not demographically different from the non participants in the pool with appropriate statistical analysis would be helpful. Currently there is a huge risk of bias in this study - of those volunteering to being interviewed and a bias for those being invited. How do the authors know that this is a representative sample of ankle fractures and is the result generalizable? How did the authors control for this? Are the results generalizable to the overall cohort - to the local region and to ankle fractures internationally as an international journal? with respect to the AIR study was the cohort representative of the age, sex, surgical management, cast and brace treatment, occupation and mechanism of injury of the total pool? How many were in the AIR study? While the inclusion and exclusion criteria of the AIR study is included the sampling of the study group of 10 is not clearly outlined. On page 5 line 48 the authors state that the sampling strategy allowed for a diverse range of patients with respect to a list of variables - but how? This is not simply because all were invited. What if the AIR study involved 1000 patients and the first 10 responding in the first recruitment were all 60 to 70 years old, female, operatively treated and placed in a brace? How was there a control over the recruitment? This is outlined in the SRQR checklist under s8. Did the authors look at the overall cohort and have a method to recruit to ensure that the interviewed cohort was representative - as it cannot be assumed. There is already an inherent bias in enrollment in an RCT.
---

	The reason I asked the number of quotes to be reduced is there is huge bias in the quotes being given such that they may represent the opinions of the authors and the message the authors want to give rather than being representative of the true experience of the patients. In the abstract the authors do not list any limitations in the study - this not a weakness to admit limitations and all studies have them. However the obvious ones have to both be listed and outlined - selection bias, bias in reporting, and limitations of generalizability because of a small study population and selection bias by both the researchers and the patients. There is no description on how this bias may have been managed or documented. Stating that the study is purposive is not enough similar to stating that a study is "case cohort" or 'randomized prospective". The study has to have a method of patient selection. So in an RCT the patient pool must be outlined, bias controlled, exactly who was included outlined, who was excluded outlined, how the sample size was determined, how the patients were invited and selected to control bias, how many were not selected, how many dropped out of the study (usually in a table) to ensure that the study remains representative of the population at large. In most journals I edit for I can make a suggestion to the editor blind to the authors with why I selected the recommendation and why I did not select reject. This journal does not offer that choice. To the editor: If these questions cannot be answered to an appropriate level to allow publication then I would recommend rejection of the paper. I hope that this can be salvaged to a level that is appropriate for publication, but the risks for bias, how they were controlled and concerns about generalizability must be addressed.
--	---

VERSION 2 – AUTHOR RESPONSE

Reviewer 1:

Comment: Methods - it is not at all clear that operative ankles are in the study. Please make this clear in inclusion criteria.

Response: Thank you for your comment on the inclusion criteria listed in table 1. I have included a line to state that the ankle fractures included here were either operatively or non-operatively managed and believe this improves the clarity of the methods.

Comment: - Patient and public involvement - names should not be used here - acknowledgement at the end only.

Response: Many thanks for your comment on the patient and public involvement section. We have removed the names of the individuals and re-written this slightly and we believe this improves the readability of the manuscript.

Comment: Results - loss of independence - 5th sentence - needs rewriting.

Response: Thanks for pointing out the error in this section, we've rewritten this and believe this has improved the quality of the results section.

Comment: References - need a good reference to return to driving eg. Archives of Orthopaedic and Trauma Surgery (2018) 138:1691–1697

<https://doi.org/10.1007/s00402-018-3037-3>

TRAUMA SURGERY

Return to car driving is safe 6 weeks after operative treatment of right ankle fractures Sean W. L. Ho¹, Michael Yam¹, M. L. Chan², Ernest B. K. Kwek¹

Response: Thanks for your comment and the useful reference suggestion. We agree this would be beneficial to include and therefore have added this in the discussion. Thank you for your help in improving this aspect of the manuscript.

Comment: Discussion - put strengths and limitations of the study at the end of the discussion. It is a bit jumbled now as it is. Rewrite this section as the end of the discussion does not fit there.

Response: Many thanks for your comment on the discussion section of our manuscript. We have moved the paragraph on strengths and limitations as suggested. We have included the conclusion straight after the discussion and hope this improves the overall quality of the manuscript.

Comment: Conclusion - add a title - Conclusion - and make a strong conclusion here. It is wimpy now. Make a strong conclusion like the abstract states eg. driving problems, loss of independence and problems of the elderly.

Response: Thank for your comments on the conclusion of the paper. We have included a title for this section and have re-written the conclusion as suggested. We hope this improves the conclusion of the manuscript.

Reviewer 2:

Comment: Reflections and corrections are satisfied

Response: Many thanks for your comment. We are pleased that you are satisfied with the amendments previously made to the manuscript.

Reviewer 3:

Comment: First - I really like the study and its message. However there is much work to be done to bring it to an appropriate standard for a journal. In particular there is no way that this study could be repeated based on the information presented. Specifically there is no way that I could repeat the study in my institution based on the information presented. Do I just walk down the hall, find 10 people with ankle fractures and bring them into a room at an assigned time and start asking questions? I need to know how to determine my sample size. If it is because I have enough budget to only interview 10 people that is fair enough - that is then the rationale for a sample size of 10. However there has to be a reason it is 10, not 1 or 100. Or is it based on the number of fractures seen in a region - and then determined that 10 is appropriate to be representative of that population.

Response: Many thanks for your appreciation of our study and its message. We have clarified the methods and results sections paying particular attention to the method of recruitment and justification of the sample size.

Comment: Once the number of participants has been determined (in studies I usually do we determine this via a power analysis to ensure that we have enough patients to know that the conclusions are valid

Response: Thanks for your comment on the fact we did not complete a sample size calculation based upon a power analysis. We agree that this technique is the most appropriate to use when conducting quantitative research methodology. However, this technique is not appropriate to use in the field of qualitative research methodology. We refer to articles within the literature which justify our approach

(1, 2). We have added further clarification to our manuscript methods and results section regarding the sample size justification.

Comment: ...the selection of participants has to be clearly outlined so that bias is controlled. Review of the demographics of the 10 patients would indicate that they are representative but a table showing that the study group is at least not demographically different from the non participants in the pool with appropriate statistical analysis would be helpful.

Response: Many thanks for your comment on the selection of participants. We selected our participants from this study using a purposive sampling framework which is described in the methods section of the manuscript. As this is a qualitative interview study, statistical analysis showing differences between the participants and non-participants is not appropriate here. Rather than seeking a representative sample, the approach in qualitative research is to sample theoretically – based on characteristics we identify as salient for the research, for example, age, gender etc. We developed a purposive sampling framework in order to achieve this.

We accept that those who declined the interview could have differed to those who accepted the interview. For example, people might have declined as they were struggling more with their recovery, or because they were doing very well and felt they had little to discuss. We have therefore included this as a limitation in the abstract and the discussion sections.

Comment: Currently there is a huge risk of bias in this study - of those volunteering to being interviewed and a bias for those being invited. How do the authors know that this is a representative sample of ankle fractures and is the result generalizable? How did the authors control for this? Are the results generalizable to the overall cohort - to the local region and to ankle fractures internationally as an international journal?

Response: Many thanks for your comment on the potential bias in this study and the generalisability of the results presented here. We have explained and expanded on this limitation in further detail in the discussion section of this manuscript. We reference an article by Marshall (3), explaining that the aim of qualitative enquiry is not for the results to be generalizable, but to gain a rich exploratory account of individuals' experiences. We hope that clarifies the issue presented here.

Comment: with respect to the AIR study was the cohort representative of the age, sex, surgical management, cast and brace treatment, occupation and mechanism of injury of the total pool? How many were in the AIR study? While the inclusion and exclusion criteria of the AIR study is included the sampling of the study group of 10 is not clearly outlined. On page 5 line 48 the authors state that the sampling strategy allowed for a diverse range of patients with respect to a list of variables - but how? This is not simply because all were invited. What if the AIR study involved 1000 patients and the first 10 responding in the first recruitment were all 60 to 70 years old, female, operatively treated and placed in a brace? How was there a control over the recruitment? This is outlined in the SRQR checklist under s8. Did the authors look at the overall cohort and have a method to recruit to ensure that the interviewed cohort was representative - as it cannot be assumed.

There is already an inherent bias in enrollment in an RCT.

Response: Many thanks for your comment regarding the AIR study sample, which we used to sample our study from. We have included details of how many were included in the AIR study at the time of data collection and how many were within the appropriate 16-23 week window in the flowchart in figure 1. We feel we have explained how we screened the patients for invitation in the first paragraph of "Recruitment and Consent" where we state that "the participants of the trial were screened for sampling attributes of age, gender, fracture management and randomised intervention." This was based on our purposive sampling framework. So as you can see, in the flowchart we show that 367 patients were in the trial and that 161 of these were eligible for invitation based upon being within the

16-23week time frame. Of these, we screened the list and invited a range of patients ensuring a diversity in these key characteristics. If the first 10 participants were all similar with regard to these variables, then they would not have all been invited to participate. We hope this clarifies our position on the sampling we used in this study.

Comment: The reason I asked the number of quotes to be reduced is there is huge bias in the quotes being given such that they may represent the opinions of the authors and the message the authors want to give rather than being representative of the true experience of the patients.

Response: Thanks for your comment on the number of quotes we have included in the results section. As the quotations included here are unedited quotations taken directly from verbatim transcribed interview records, we are confident that they are representative of the patient's experiences as the respondents expressed them throughout the interview process. We have chosen quotes which best articulate our themes and categories, after a rigorous thematic analysis.

In our manuscript we have used 38 quotations. We have reviewed six studies of a similar nature (4-9) and the number of quotations used ranges from 20 to 38. Our manuscript is therefore consistent with these articles. We feel that removing any quotations may introduce further bias and the more quotations in the manuscript, the closer we are to representing the views, opinions and experiences of patients correctly and accurately. We thank you for your suggestion and comment on our manuscript.

Comment: In the abstract the authors do not list any limitations in the study - this not a weakness to admit limitations and all studies have them. However the obvious ones have to both be listed and outlined - selection bias, bias in reporting, and limitations of generalizability because of a small study population and selection bias by both the researchers and the patients.

Response: Thanks for your comment on our strengths and limitations section in the abstract. We would like to point out the limitation we included in the abstract, in bullet point 4 of the strengths and limitations section. This point refers to the clinical trial being a limiting factor of the study process because it restricted the timing of the interviews. We have included a further bullet point related to a limitation in the study and we hope that improves this section of the manuscript.

Comment: There is no description on how this bias may have been managed or documented. Stating that the study is purposive is not enough similar to stating that a study is "case cohort" or "randomized prospective". The study has to have a method of patient selection. So in an RCT the patient pool must be outlined, bias controlled, exactly who was included outlined, who was excluded outlined, how the sample size was determined, how the patients were invited and selected to control bias, how many were not selected, how many dropped out of the study (usually in a table) to ensure that the study remains representative of the population at large.

Response: Many thanks for your comment on the manuscript. To clarify we categorised this article as a qualitative interview study which utilised a purposive sampling strategy. We appreciate that in quantitative studies, such as randomised controlled trials, there are very different ways of detailing how the patients are identified and recruited. Again, we would like to draw attention to the flowchart shown in figure 1 which details how many participants were eligible for inclusion in the study, how many were invited, how many did not participate and the reasons for this. We would like to further remind the reviewer that the goal of qualitative research is not for the results to be generalizable across the population as a whole, but rather create a rich exploratory account of experiences to inform further enquiry in the research area. We acknowledge that there is likely to be bias present within the study, which we have explained in the discussion, but accept that the researchers are an integral part of the qualitative research process. We have explained how we used reflexivity to ensure

that we regularly reflected on and adjusted our position within the enquiry throughout the iterative process of the research. This process is outlined and explained further by Marshall (3).

References

1. Guest G, Bunce A, Johnson L. How Many Interviews Are Enough?: An Experiment with Data Saturation and Variability. *Field Methods*. 2006;18(1):59-82.
2. Saunders B, Sim J, Kingstone T, Baker S, Waterfield J, Bartlam B, et al. Saturation in qualitative research: exploring its conceptualization and operationalization. *Quality & Quantity*. 2018;52(4):1893-907.
3. Marshall MN. Sampling for qualitative research. *Family Practice*. 1996;13(6):522-6.
4. Griffiths F, Mason V, Boardman F, Dennick K, Haywood K, Achten J, et al. Evaluating recovery following hip fracture: a qualitative interview study of what is important to patients. *BMJ Open*. 2015;5(1):e005406.
5. Hallberg I, Ek A-C, Toss G, Bachrach-Lindström M. A striving for independence: a qualitative study of women living with vertebral fracture. *BMC Nursing*. 2010;9(1):7.
6. Rees S, Tutton E, Achten J, Bruce J, Costa ML. Patient experience of long-term recovery after open fracture of the lower limb: a qualitative study using interviews in a community setting. *BMJ Open*. 2019;9(10):e031261.
7. McPhail S, Dunstan J, Canning J, Haines T. Life impact of ankle fractures: Qualitative analysis of patient and clinician experiences. *BMC Musculoskeletal Disorders*. 2012;13.
8. Pinsker EB, Sale JEM, Gignac MAM, Daniels TR, Beaton DE. "I don't have to think about watching the ground": a qualitative study exploring the concept of vigilance as an important outcome for ankle reconstruction. *Arthritis Care & Research*. 2019;0(ja).
9. Archibald G. Patients' experiences of hip fracture. *Journal of Advanced Nursing*. 2003;44(4):385-92.

VERSION 3 – REVIEW

REVIEWER	Dr. Richard Buckley Foothills Medical Center University of Calgary, Canada
REVIEW RETURNED	12-Dec-2019

GENERAL COMMENTS	Very good revision. In one section the term "female" physiotherapist is used. This need not be advertised. It should be removed.
---